# Effects of *Scutellaria baicalensis* Extract-Induced Exosomes on the Periodontal Stem Cells and Immune Cells under Fine Dust

**DOI:** 10.3390/nano14171396

**Published:** 2024-08-27

**Authors:** Mihae Yun, Boyong Kim

**Affiliations:** 1Department of Dental Hygiene, Andong Science College, Andong-si 36616, Republic of Korea; ymh0710@asc.ac.kr; 2EVERBIO, 131, Jukhyeon-gil, Gwanghyewon-myeon, Jincheon-gun 27809, Republic of Korea

**Keywords:** periodontal ligament cell, pulp progenitor cell, osteogenesis, extracellular vesicle, microRNA

## Abstract

In adverse environments, fine dust is linked to a variety of health disorders, including cancers, cardiovascular, neurological, renal, reproductive, motor, systemic, and respiratory diseases. Although PM10 is associated with oral inflammation and cancer, there is limited research on biomaterials that prevent damage caused by fine dust. In this study, we evaluated the effects of biomaterials using microRNA profiling, flow cytometry, conventional PCR, immunocytochemistry, Alizarin O staining, and ELISA. Compared to SBE (*Scutellaria baicalensis* extract), the preventive effectiveness of SBEIEs (SBE-induced exosomes) against fine dust was approximately two times higher. Furthermore, SBEIEs promoted cellular differentiation of periodontal ligament stem cells (PDLSCs) into osteoblasts, periodontal ligament cells (PDLCs), and pulp progenitor cells (PPCs), enhancing immune modulation for oral health against fine dust. In terms of immune modulation, SBEIEs activated the secretion of cytokines such as IL-10, LL-37, and TGF-β in T cells, B cells, and macrophages, while attenuating the secretion of MCP-1 in macrophages. MicroRNA profiling revealed that significantly modulated miRNAs in SBEIEs influenced four biochemical categories: apoptosis, cellular differentiation, immune activation, and anti-inflammation. These findings suggest that SBEIEs are an optimal biomaterial for developing oral health care products. Additionally, this study proposes functional microRNA candidates for the development of pharmaceutical liposomes.

## 1. Introduction

*Scutellaria baicalensis* is a flowering plant and is native to Korea, China, Mongolia, and Russia [1]. The root of the plant contains various phytochemicals, including baicalein, baicalin, wogonin, nor-wogonin, oroxylin A, and β-sitosterol [2]. According to [3], the root shows various biological functions, including anti-tumor, anti-bacterial, anti-viral, antioxidant, and neuroprotective effects. Based on the biological functions of these compounds, extensive research has demonstrated that *Scutellaria baicalensis* extract (SBE) possesses several benefits for dental care, including the suppression of periodontitis [4], enhancement of osteogenic differentiation [5], and regulation of pro-inflammatory responses and innate immunity [6]. However, despite baicalin being one of the most efficacious compounds within SBE, it exhibits low bioavailability in vivo [7]. The efficacy of other compounds within SBE is insufficient for development into pharmaceutical materials [7]. Furthermore, prolonged use of SBE induces elevation of blood glucose and liver injury [8].

MicroRNAs (miRNAs) transcribed by RNA polymerase are small noncoding RNA molecules containing 21 to 23 nucleotides [9]. Under stimuli, ordinary cells modulate their levels of miRNAs and transfer miRNAs to other cells through exosome-shutting [10]. The transferred miRNAs modulate translation of mRNAs in target cells [11]. Although in recent times stem cell-based therapy has shown considerable potential against various diseases, the therapy comes with inescapable drawbacks, including immunogenicity, infusion toxicity, effort preservation, ethical issues, and tumorigenic potential [12]. Contrary to the drawbacks of this therapy, cell-induced exosomes are free from these drawbacks, and they show a stronger effect than the therapy [12]. Furthermore, induced exosomes from stem cells are applicable to various surgical fields, including obstetrics, gynecological, orthopedic, plastic, cardiothoracic, urologic, and ophthalmologic surgeries [12]. Profiling of significant miRNAs in exosomes is crucial to develop more effective biomaterials, such as a liposome and a hybrid between a liposome and an exosome [13]. 

Fine dust, commonly referred to as particulate matter, is prevalent in the atmosphere and includes particles with diameters of 2.5 μm (PM2.5) to 10 μm (PM10), which pose significant health risks to humans and animals [14]. Particularly vulnerable populations include the elderly, children, and individuals with respiratory conditions [15,16]. Exposure to fine dust is linked to a variety of health disorders, such as cardiovascular, neurological, renal, reproductive, motor, systemic, and respiratory diseases, and various cancers [17,18]. The presence of pathogenic bacteria and viruses in PM10 exacerbates pathogenic infections within the oral cavity [19,20,21]. Additionally, PM10 is associated with oral inflammation and cancer, as well as multiple other types of cancer throughout the body [22]. Gingival inflammation, triggered by fine dust, promotes bacterial platelet aggregation, resulting in increased levels of C-reactive proteins and amyloid A fibrinogens in the liver, and contributes to atherosclerosis in humans [23]. Recent research indicates that fine dust inhibits the osteogenic differentiation of adipose-derived stem cells [24]. Furthermore, fine dust exposure induces inflammation in dermal tissues in both humans and pets [23]. This skin inflammation compromises dermal immunity, affecting the differentiation of adipose-derived stem cells (ASCs) in subcutaneous fat tissues [24,25]. Upon exposure to fine dust, dermal cells exhibit upregulation of apoptotic proteins, such as BAX and CytC, and downregulation of antiapoptotic proteins, including AKT, P50, P52, and BCL-2 [24].

In the past 20 years, the frequency of dental disorders has shown a significant increase [26]. In addition to treating periodontitis, dental implants have become critical in oral healthcare. However, there are currently no materials that have effectively improved the success rate of these surgeries. Therefore, the development of functional biomaterials is imperative to reduce side effects and enhance surgical outcomes. This study proposes a biofunctional material that modulates interactions between periodontal cells and immune cells within the periodontium. It also activates the differentiation of periodontal ligament stem cells (PDLSCs) to osteoblasts, periodontal ligament cells (PDLCs), and pulp progenitor cells (PPCs), despite exposure to fine dust.

## 2. Materials and Methods

### 2.1. Cell Culture and Cytotoxicity Test for Establishing Treating Dosage 

Human normal gingival cells (PCS-201-018, ATCC, Manassas, VA, USA) were cultured in their complete growth media kits (PCS-201-030 and PCS-201-041, ATCC) at 37 °C and 5% CO_2_. To isolate induced exosomes, the cultured gingival cells were exposed to 400ug/mL *Scutellaria baicalensis* extract (SBE), made using 50% ethanol and fine dust (PM10) (ERM-CZ100, Sigma-Aldrich, St. Louis, MO, USA) for one day. To isolate induced exosomes, the supernatants were collected from gingival cells under various conditions (Con, SBE, PM10). The induced exosomes (CIE, SBEIE, PM10IE) were isolated and purified from the supernatants (10mL) using the exoEasy Maxi Kit (QIAGEN, Hilden, Germany) and CD68 Exo-Flow Capture Kit (System Biosciences, Palo Alto, CA, USA) respectively. Additionally, the cultured periodontal ligament stem cells (PDLSCs) (SKU: 36085-01and M36085-01S, Celprogen, Torrance, CA, USA) were exposed to SBE-conditioned medium (SBECM) along with the induced exosomes (CIE; control-induced medium, PM10IE; PM10-induced medium, SBEIE; SBE-induced exosomes) to establish their treatment dosages. To establish the treatment dosages, macrophage (KG1, ATCC), B (SKW6.4, ATCC), and T cells (Jurkat, ATCC) were exposed to the induced exosomes for one day. Their established dosages are provided in the Appendix A. The cytotoxicity concentrations were evaluated with using a flow cytometer (BD FACScalibur, BD Biosciences, San Jose, CA, USA) and FlowJo 10.6.1 (BD Biosciences).

### 2.2. Analysis for Differentiating Patterns of PDLSCs under the Biomaterials 

After the PDLSCs were cultured under various conditions (Con, PM10CM, SBECM, and SBECM + PM10), the induced exosomes (CIE, PM10IE, SBEIE, SBEIE + PM10IE) and then the cultured cells were fixed with 2% paraformaldehyde for 4 h and treated with 0.02% Tween 20 for 5 min. After blocking with Fc blocker reagent (BD Biosciences), the treated cells were incubated with three fluorescence-conjugated immunoglobulins, FITC-anti-asporin (Abbexa, Cambridge, UK), PE-anti- osteopontin (R&D Systems, Minneapolis, MN, USA), and APC-anti- cytokeratin 4 (Biorbyt, Cambridge, UK), at 37 °C for two days. The stained cells were evaluated using a flow cytometer (BD FACScalibur), FlowJo 10.6.1 (BD science), and Prism 7 (GraphPad, San Diego, CA, USA). 

### 2.3. Localization of PLDC Marker Using Immunocytochemistry

After the periodontal ligament stem cells (PDLSCs) were cultured under various conditions (CIE, PM10IE, SBEIE, SBEIE + PM10IE), the cultured cells were fixed with 2% paraformaldehyde for 12 h and treated with 0.02% Tween 20 for 10 min. After blocking with Fc blocker reagent (BD Biosciences), the treated cells were incubated with three fluorescence-conjugated immunoglobulins, FITC-anti-asporin (Abbexa, Cambridge, UK), for two days. The stained cells were evaluated using a fluorescence microscope (Eclipse Ts-2, Nikon, Shinagawa, Japan) and the imaging software NIS-elements V5.11 (Nikon).

### 2.4. Profiling of microRNAs in SBEIE

The isolated and purified exosomes were sequenced by ebiogen Inc. (Seoul, Republic of Korea) to analyze exosomal functions. An Agilent 2100 bio-analyzer and the RNA 6000PicoChip (Agilent Technologies, Amstelveen, The Netherlands) were used to evaluate RNAquality. RNA was quantified using a NanoDrop 2000 spectrophotometer (Thermo FisherScientific, Waltham, MA, USA). Small RNA libraries were prepared and sequenced using the Agilent 2100 Bio-analyzer instrument for the high-sensitivity DNA assay (Agilent Technologies, Inc., Santa Clara, CA, USA) and the NextSeq500system for single-end 75 sequencing (Illumina, San Diego, CA, USA). To obtain an alignment file, the sequences were mapped using bowtie 2 software (CGE Risk, Lange Vijverberg, The Netherlands), and the read counts were extracted from the alignment file using bedtools (v2.25.0) (GitHub, Inc., San Francisco, CA, USA) and R language (version 3.2.2) (R studio, Boston, MA, USA) to evaluate the miRNA expression level. miRWalk 2.0 (Ruprecht-Karls-Universität Heidelberg, Medizinische Fakultät Mannheim, Germany) was used for the miRNA target signal study, and ExDEGA v.2.0 (ebiogen Inc., Seoul, Republic of Korea) was used to deduce radar charts.

### 2.5. Evaluating for the Levels of Osteogenic Markers 

Total RNAs in cultured cells under various conditions were isolated from the treated cells using RiboEx reagent (GeneAll, Seoul, Republic of Korea), and cDNA was synthesized from the isolated RNA using Maxime RT PreMix (iNtRON, Seongnam, Republic of Korea). The cDNA was amplified with primers (Table 1) under the following cycling parameters: 1 min at 95 °C, followed by 35 cycles of 35 s at 59 °C, and 1 min at 72 °C. The amplified DNA was estimated using iBright FL1000 and iBright Analysis Software 4.0.0 (Invitrogen, Waltham, MA, USA). 

### 2.6. Imagine Analysis of Osteoblast Cells

After the periodontal ligament stem cells (PDLSCs) were cultured under various conditions with the two materials (SBE and induced exosomes), the cultured cells were fixed with 2% paraformaldehyde for 12 h and stained using Alizarin O reagent (Sigma, St. Louis, MO, USA) for 40 min. The stained cells were evaluated using a fluorescence microscope (Eclipse Ts-2, Nikon, Shinagawa, Japan) and the imaging software NIS-elements V5.11 (Nikon).

### 2.7. Evaluating for Bacterial and Viral Phagocytosis

Macrophages (KG1, ATCC) were cultured under various conditions (CIE, SBECM, SBECM + PM10, PM10IE, SBEIE, and SBEIE + PM10IE). The exposed cells were treated with bacterial particles, BioParticles (Thermo Fisher Scientific, Waltham, MA, USA), and human papilloma viral peptide (synthesized FITC-conjugated HPV16 E7(83–97), LMGTLGIVCPICSQK). The treated cells were evaluated using a flow cytometer (BD FACScalibur), FlowJo 10.6.1 (BD bioscience), and Prism 7 (GraphPad, San Diego, CA, USA).

### 2.8. Evaluating of Cytokines in Immune Cells 

After the cultured macrophages, T cells, and B cell were exposed to various conditions (CIE, SBECM, SBECM + PM10, PM10IE, SBEIE, and SBEIE + PM10IE) for one day, their culture media were isolated. Cytokines in the isolated media were evaluated using IL-10 (Interlekin-10) (Thermo Fisher Scientific), TGF-β (Transforming growth factor-β) (Abcam, Cambridge, UK), LL-37 (Novus Biologicals, Centennial, CO, USA), and MCP-1 (Monocyte Chemoattractant Protein 1) (Thermo Fisher Scientific) ELISA kits and a microplate reader (AMR-100; Allsheng, Hangzhou, China).

### 2.9. Statistical Analysis 

All experiments were analyzed using one way analysis of variance (ANOVA) with post hoc (Scheffe’s method) using Prism 7 software (GraphPad, San Diego, CA, USA).

## 3. Results

In this manuscript, SBE-induced exosomes reveal two biofunctions, including activation of cellular differentiation and modulation of immunity among various types of immune cells. Based on the results of cytotoxicity (Figure 1), treatment dosages were established as 400 μg/mL of SBE and 16 μg/mL of PM10 for gingival cells and 60 μg/mL of CIE, 30 μg/mL of SBEIE, and 40 μg/mL of PM10IE for PDLSCs (Figure 1). 

In the results for profiling of microRNAs in SBEIE (Figure 2 and Table 2), regulated miRNAs affected three biological categories, including apoptosis, cellular differentiation, and immunity (Figure 2). Compared to miRNAs of PM10IE, hsa-miR-151a-3p and hsa-miR-140-3p showed dramatic upregulation in SBEIE and modulated the three categories (Figure 2 and Table 2). Notably, corresponding to other results in this manuscript, SBEIE demonstrated strong upregulation of miRNAs associated with cellular differentiation and immune modulation (Figure 2 and Table 2). These values of miRNAs in SBEIE were more than four times other normalized miRNAs. Among the most dramatically upregulated miRNAs, hsa-miR-148a-3p and hsa-miR-378a-3p affect only the differentiation and immune activation categories (Table 2).

### 3.1. Activation for Differentiation of PDLSCs by SBEIE

The purpose of this experiment (Figure 3) is to evaluate the cell differentiation activity of the supernatants derived from cells stimulated by natural products, which contain not only induced exosomes but also various other substances. By assessing the differentiation potential of these supernatants and comparing it with the activity of induced exosomes, this study aims to provide foundational data for evaluating the potential of SBEIE as a bioactive material. This comparative analysis will serve as a basis for exploring the biotechnological applications of SBEIE. Supernatants from gingival cells treated with SBE (SBECM) activated the differentiation of PDLSCs into osteoblasts, PDLCs, and PPCs, despite exposure to fine dust. Notably, SBECM significantly enhanced PDLC differentiation, with values 50.7 times higher than those observed with PM10CM (Figure 3). Remarkably, SBEIE greatly reinforced differentiation, with the relative values 1290 times higher than those of PM10CM (Figure 4). Compared to SBECM, SBEIE more effectively activated differentiation, with relative values approximately 25.4 times higher (Figure 3 and Figure 4).

Although SBEIE was effective for PDLC differentiation in terms of its preventive effect against fine dust, compared to SBECM, SBEIE was more effective for osteogenic differentiation (Figure 3 and Figure 4). The relative values were 2.6 times higher than those of SBECM (Figure 3 and Figure 4).

Based on the results of immunocytochemistry (Figure 5), the effects of SBEIE correspond to the results obtained from flow cytometry (Figure 3 and Figure 4). SBEIE activated the expression of asporin, a marker of PDLCs, with values 9.4 times higher compared to those of PM10IE (Figure 5).

Additionally, SBEIE upregulated osteogenic markers, including *BMP2, Runx2,* and *Dlx5* in PDLSCs, despite exposure to fine dust (Figure 6). Compared to SBECM, SBEIE significantly upregulated Runx2, with values approximately 2.33 times higher under fine dust conditions (Figure 6a). Similarly, the results for the osteogenic markers (Figure 6a) were corroborated by the outcomes of Alizarin O staining under various conditions (Figure 6b).

### 3.2. Activation of Immune System

SBEIE modulated immune activities and protected immune cells against fine dust (Figure 7 and Figure 8). First, SBEIE activated phagocytic activity in macrophages exposed to fine dust (Figure 7). Compared to PM10IE, SBEIE strongly enhanced phagocytosis against bacterial particles and viral peptides (Figure 7a,b). Although phagocytic activities were attenuated by approximately 50% under fine dust conditions, SBEIE increased these activities approximately 9.14 times (Figure 7a,b). Furthermore, SBEIE modulated the secretion of cytokines, including IL-10, TGF-β, LL-37, and MCP-1, in B cells, T cells, and macrophages. SBEIE upregulated IL-10 approximately 9.87 and 5.3 times more than PM10IE in B cells and T cells, respectively, (Figure 8). Notably, SBEIE intensely upregulated TGF-β in T cells under fine dust exposure (Figure 8). In contrast to these results, SBEIE suppressed the expression of MCP-1 in macrophages. Quantitatively, the concentrations of cytokines under SBEIE treatment were 120 pg/mL of IL-10 in B cells, 80 pg/mL of IL-10 in T cells, 4 ng/mL of TGF-β in T cells, and 20 pg/mL of MCP-1 in macrophages (Figure 8). In regulation of LL-37, SBEIE dramatically activated upregulation of LL37 compared to PM10IE (Figure 8). SBEIE was 7.3 times more than PM10IE (Figure 8).

## 4. Discussion

Many studies have reported biofunctional phytochemicals associated with various diseases, and several extraction methods have been developed to improve the production of health-functional items in industries [27]. However, there are numerous limitations, including the high cost of purification, potential side effects, and low commercial benefits from the purification and commercialization of these compounds [28]. In this study, we propose the use of functional biomaterials, specifically exosomes derived from normal gingival cells, which exhibit biological functions such as promoting differentiation, enhancing immunity, and preventing the adverse effects of fine dust in the periodontium. 

SBEIE activates various biological functions in two biochemical categories, including cellular differentiation and immune modulation. 

First, in cellular differentiation, SBEIE activated differentiation of PDLSCs in three types of cells, including osteoblasts, PDLCs, and PPCs under fine dust (Figure 3, Figure 4, Figure 5 and Figure 6). Compared to PM10IE, SBEIE contained six dramatically upregulated miRNAs (Table 2). Among the six miRNAs, two miRNAs (hsa-miR-151a-3p, hsa-miR-140-3p) suppressed cellular apoptosis. Corresponding to these profiles, PDLSCs exposed to SBEIE intensely differentiated the three types of cells. Notably, SBEIE displayed the most effective protection for PDLC differentiation among the differentiation of three types of the cells under fine dust (Figure 3 and Figure 4). However, compared to SBECM, SBEIE was more effective for osteogenic differentiation. These results suggest that the six miRNAs in SBEIE activate differentiation to PDLCs besides suppression of apoptosis by fine dust. Based on the trend in dental implant procedures [29], 100,000 to 300,000 dental implants are placed annually. Osseointegration of the dental fixture is crucial for successful implantation [30]. Given the limited number of osseointegration catalysts for dental fixtures, the development of new materials is necessary. In recent reports [31,32], hsa-miR-148a-3p activates differentiation of skeletal muscle, endothelial cells, adipocytes, and osteoblasts. SBEIE and liposomes with functional miRNA candidates (Table 2) are proposed as optimal osseointegration catalysts in the dental industry. 

Second, SBEIE modulated immune activity among B cells, T cells and macrophages to maintain periodontal health under fine dust. According to the recent reports [33,34], various immune cells, including CD4^+^ T cells, CD8^+^ T cells, Treg cells, and B1 cells, secrete IL-10, IL-17, and TGF-β to maintain periodontal health [33]. Contrary to that condition, periodontal diseases induce suppression of Treg cells and the activation of a RANK (Receptor activator of nuclear factor kappa-Β) signal in B cells [33]. SBEIE modulated secretion of IL-10, TGF-β, and MCP-1 in immune cells, including B cells, T cells, and macrophages (Figure 8). IL-10 and TGF-β activate the maintenance of periodontal homeostasis, suppression of inflammation and osteoclastogenesis, and down-regulation of inflammatory responses in dental environments [33]. Intense upregulation of IL-10 and TGF-β by SBEIE suggests that SBEIE reinforces periodontal health despite exposure to fine dust. Additionally, the profiling results for miRNAs in SBEIE (Figure 2 and Table 2) support these functions. According to the reports, miRNAs hsa-miR-148a-3p and hsa-miR-378a-3p suppress cancer activities, chronic inflammation, and autoimmune diseases [35,36]. Notably, hsa-miR-148a-3p influences T cell receptor signaling and cytokine production and suppresses pro-inflammatory pathways, potentially contributing to an anti-inflammatory macrophage phenotype [37,38]. The significant upregulation of miRNAs, specifically hsa-miR-148a-3p and hsa-miR-378a-3p, in SBEIE plays a crucial role in immune modulation of B and T cells. 

Ordinarly, MCP-1 induces activation and migration of leukocytes under inflammation [39,40]. Under invasion of pathogens in the body, the immune system upregulates this molecule, and the molecule enhances elevation of other inflammatory cytokines and the proliferation of inflammatory cells [40]. Additionally, compared to a healthy condition, MCP-1 levels are higher in a gingival crevicular fluid (GCF) and serum in in subjects with chronic periodontitis [41]. Although MCP-1 plays a crucial role post-inflammation, elevated MCP-1 in pre-inflammation accelerates periodontitis by fine dust. The suppression of MCP-1 in macrophages by SBEIE (Figure 8) suggests that SBEIE is a very effective material for preventing periodontitis caused by fine dust. 

LL-37, a cathelicidin-derived antimicrobial peptide, exhibits significant antimicrobial activity against both gram-positive and gram-negative bacteria [42]. Additionally, various reports [43,44,45] have highlighted LL-37’s multifaceted functions, including anticancer activity, tissue healing, wound care, suppression of biofilm formation, inflammation regulation, and protection against pathogenic infections. In the oral environment, LL-37 plays several critical roles, such as antimicrobial action, immunomodulation, suppression of autoimmune diseases and carcinomas, and maintenance of homeostasis [46] hsa-miR-378a-3p has demonstrated significant antimicrobial activity, particularly through its interaction with the antimicrobial peptide LL37 [47]. Additionally, hsa-miR-378a-3p can modulate immune responses by influencing the expression of IL-33, a cytokine involved in inflammatory responses [47]. The upregulation of LL-37 by SBEIE suggests that SBEIE is an optimal biomaterial for developing various oral health care products. Furthermore, research into liposomes containing miRNA candidates is necessary for potential applications in the pharmaceutical industry.

## 5. Conclusions

This study highlights the significant potential of SBEIE in addressing the negative impacts of fine dust exposure on oral health. SBEIE effectively promotes the differentiation of periodontal ligament stem cells (PDLSCs) into osteoblasts, periodontal ligament cells (PDLCs), and perivascular progenitor cells (PPCs), and demonstrates immune-modulating effects. Additionally, profiling functional microRNAs (miRNAs) in SBEIE provides further insights into its mechanisms of action. In the realm of oral health, SBEIE offers promising applications. It could be developed into therapeutic agents for periodontal regeneration, aiding in the repair and regeneration of periodontal tissues and enhancing clinical outcomes for patients with periodontal disease. Its capacity to support bone regeneration positions SBEIE as a potential component in biomaterials designed for bone repair, particularly in cases of oral bone loss. Furthermore, SBEIE’s immune-modulating properties suggest its utility in protecting oral mucosa from pollutants such as fine dust, potentially leading to innovative treatments for conditions aggravated by environmental exposure. The ability of SBEIE to enhance tissue regeneration and modulate cellular responses also opens opportunities for personalized oral health products. While the findings are based on in vitro studies, they lay a strong foundation for future research. This study presents preliminary results, which necessitate further research to determine the optimal concentration and to investigate potential side effects through animal or clinical trials. Should future in vivo experiments demonstrate positive outcomes, it could pave the way for the development of a novel biopharmaceutical material.

## Figures and Tables

**Figure 1 nanomaterials-14-01396-f001:**
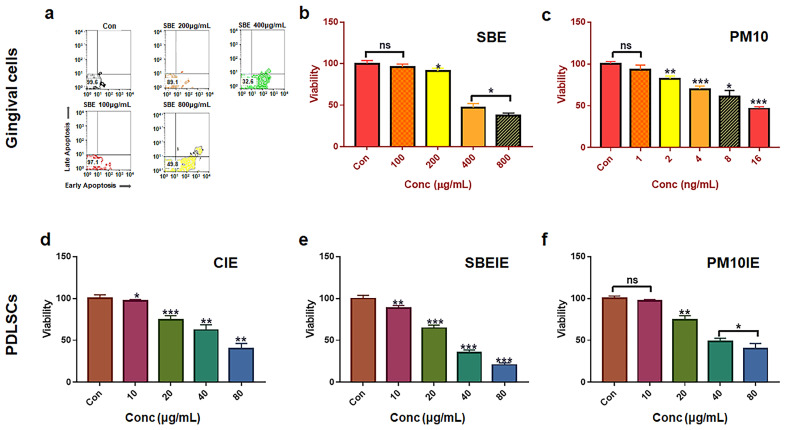
Establishment of treatment dosages for gingival cells and PDLSCs. Cellular viability of SBE (**a**,**b**), fine dust (PM10) (**c**), and exosomes from gingival cells under various conditions; control-induced exosomes (**d**), SBE-induced exosomes (**e**), and PM10-induced exosomes (**f**) using flow cytometry; ns: not significant (* *p* < 0.05, ** *p* < 0.01, *** *p* < 0.001).

**Figure 2 nanomaterials-14-01396-f002:**
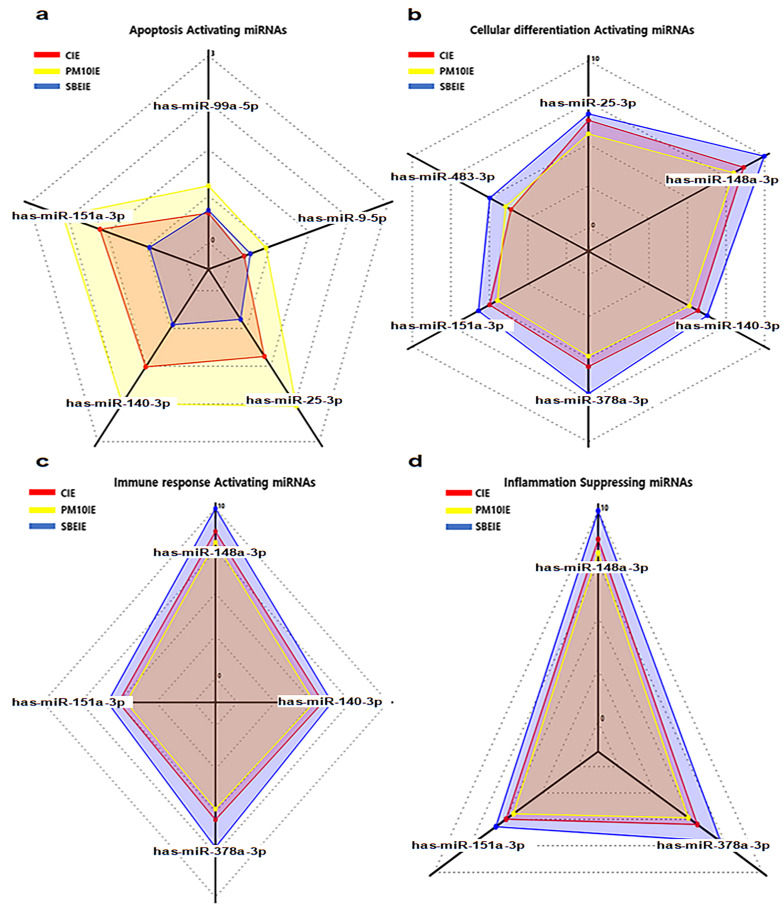
Profiling of microRNAs in induced exosomes from gingival cells. Radar charts for alteration of miRNAs in induced exosomes. CIE: control-induced exosomes; PM10IE: PM10-induced exosomes; SBEIE: SBE-induced exosomes; *p* < 0.05.

**Figure 3 nanomaterials-14-01396-f003:**
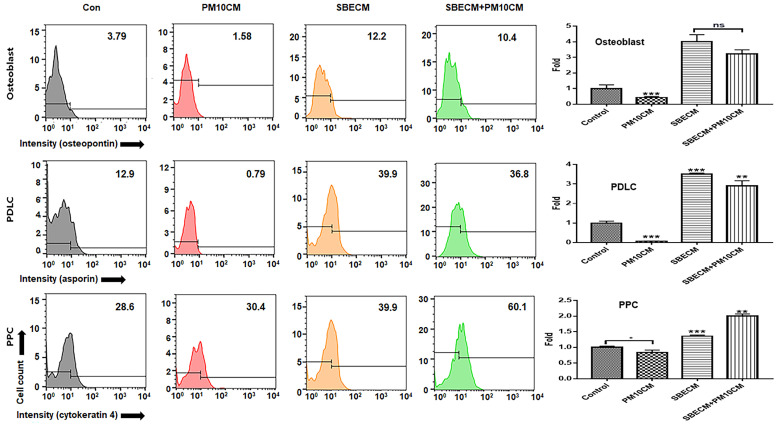
Differentiation of PDLSCs under conditioned media. Patterns of differentiation from PDLSCs under the conditioned media (con, PM10CM, SBECM, and SBECM + PM10). Con: control; PM10CM: supernatant from gingival cells under PM10; SBECM: supernatant from gingival cells under SBE; SBECM + PM10: supernatant from gingival cells sequential exposed to under SBECM and PM10; PDLC: periodontal ligament cells; PPC: pulp progenitor cells; ns: not significant (* *p* < 0.05, ** *p* < 0.01, *** *p* < 0.001).

**Figure 4 nanomaterials-14-01396-f004:**
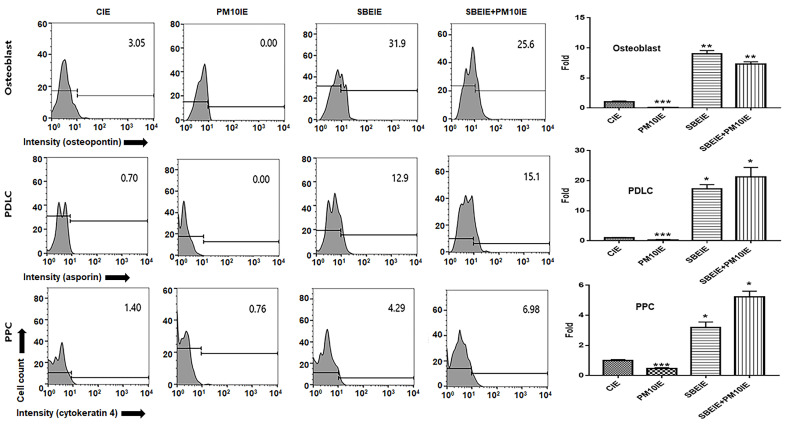
Differentiation of PDLSCs under induced exosomes. Patterns of differentiation from PDLSCs under the conditioned media (CE, PM10IE, SBEIE, and SBEIE + PM10). CE: control-induced exosomes; PM10IE: PM10-induced exosomes; SBEIE: SBE-induced exosomes; SBEIE + PM10: sequential exposure under SBEIE and PM10; PDLC: periodontal ligament cells; PPC: pulp progenitor cells (* *p* < 0.05, ** *p* < 0.01, *** *p* < 0.001).

**Figure 5 nanomaterials-14-01396-f005:**
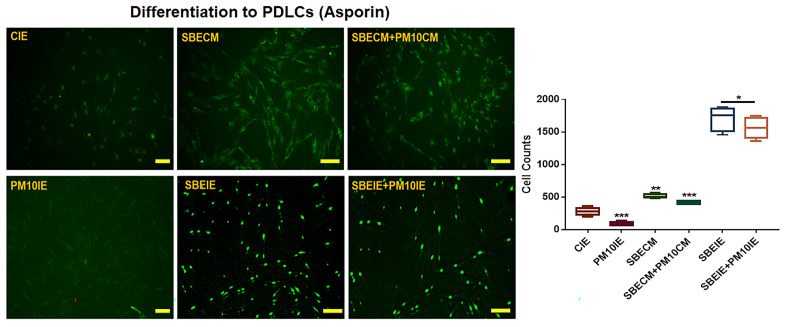
PDLC differentiation from PDLSCs under induced exosomes. Images for differentiated PDLCs using immunocytochemistry. Asporin-expressed cells show as green. Differentiation from PDLSCs under the conditioned media (CE, PM10IE, SBEIE, and SBEIE + PM10). CE: control-induced exosomes; PM10IE: PM10-induced exosomes; SBEIE: SBE-induced exosomes; SBEIE + PM10: sequential exposure under SBEIE and PM10; PDLC: periodontal ligament cells; PPC: pulp progenitor cells; ns: not significant; scale bars 10 μm (* *p* < 0.05, ** *p* < 0.01, *** *p* < 0.001).

**Figure 6 nanomaterials-14-01396-f006:**
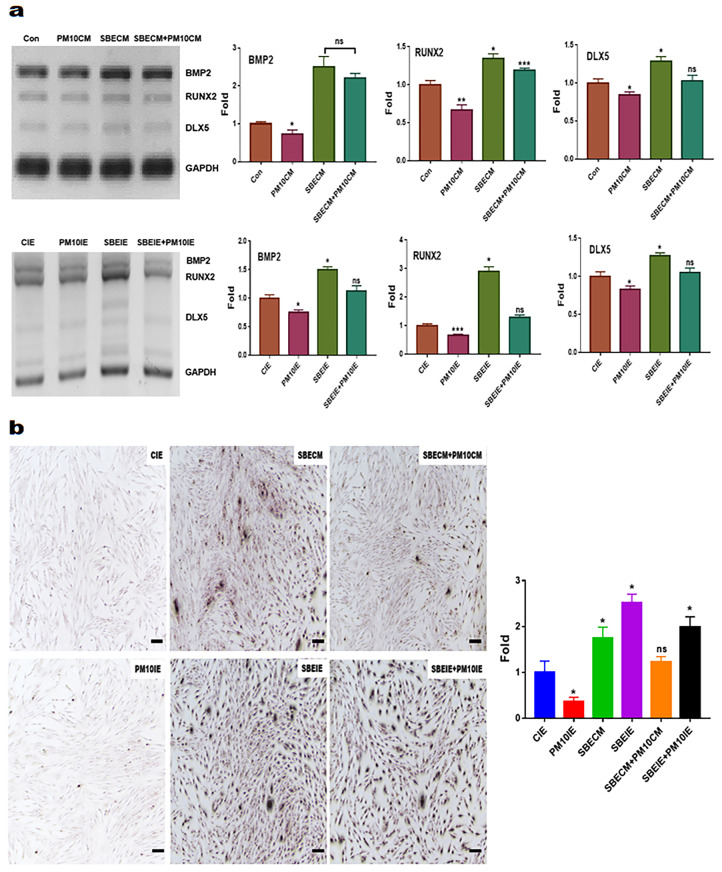
Osteogenic differentiation from PDLSCs under induced exosomes. Levels of osteogenic markers under induced exosomes (**a**). Results of Alizarin O stain for differentiated cells under induced exosomes and the bar graph display the relative folds for cell counts (**b**). ns: not significant; scale bars 10 μm (* *p* < 0.05, ** *p* < 0.01, *** *p* < 0.001).

**Figure 7 nanomaterials-14-01396-f007:**
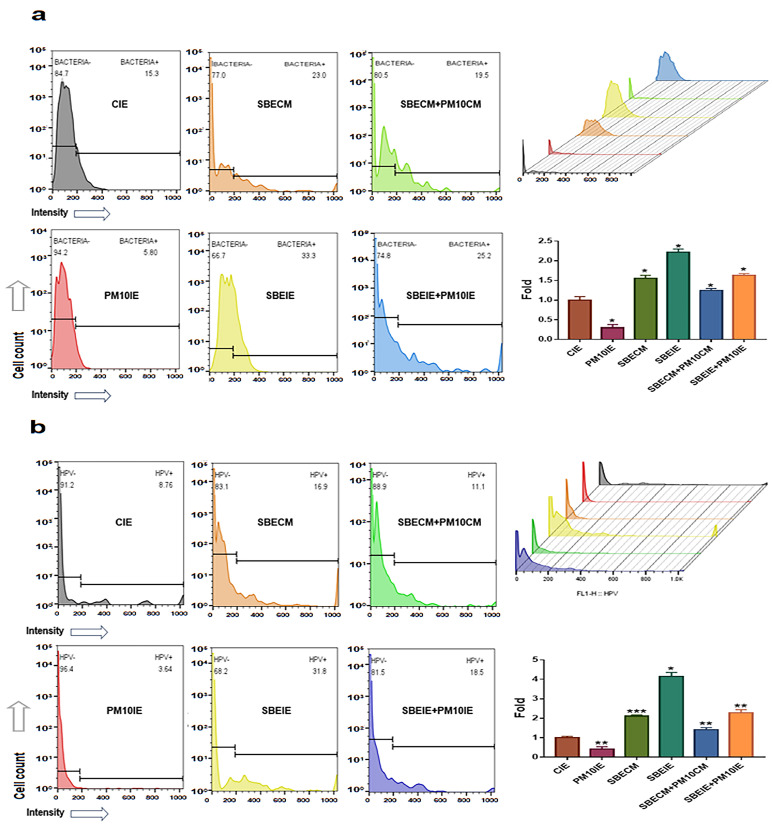
Phagocytic activation of macrophages under induced exosomes. Phagocytic activation of macrophages for bacteria (**a**) and FITC labelled viral peptides (**b**) under induced exosomes (* *p* < 0.05, ** *p* < 0.01, *** *p* < 0.001).

**Figure 8 nanomaterials-14-01396-f008:**
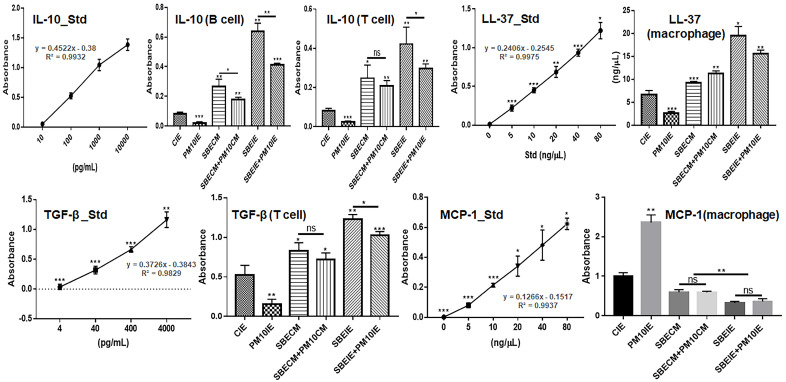
Enhancement of periodontal health by induced exosomes. Expression of IL-10 in B and T cells, TGF-β in T cells, and LL-37 and MCP-1 in macrophages under induced exosomes. The levels of cytokines evaluate using ELISA. ns: not significant (* *p* < 0.05, ** *p* < 0.01, *** *p* < 0.001).

**Table 1 nanomaterials-14-01396-t001:** Primer design for PCR.

Gene	Sequence (5′→3′)
*GAPDH*	Forward primer GGTCACCAGGGCTGCTTTTA Reverse primer CCCGTTCTCAGCCATGTAGT
*DLX2*	Forward primer CTGCTTAGACCAGAGCAGCC Reverse primer CTGGAACGGAGCTTGGAAGT
*RNX2*	Forward primer CGCCTCACAAACAACCACAG Reverse primer TCACTGTGCTGAAGAGGCTG
*BMP2*	Forward primer CTGAAACAGAGACCCACCCC Reverse primer TGGTCACGGGGAATTTCGAG

**Table 2 nanomaterials-14-01396-t002:** Profiling of microRNAs in induced exosomes.

Category	Apoptotic Activation	Differentiating Activation	Immune Activation	Anti-Inflammation
Dramatic Up-regulation	hsa-miR-151a-3p hsa-miR-140-3p hsa-miR-25-5p hsa-miR-99a-5p hsa-miR-9-5p	hsa-miR-183-3p hsa-miR-151a-3p hsa-miR-376a-3p hsa-miR-140-3p hsa-miR-148a-3p hsa-miR-25-3p	hsa-miR-148a-3p hsa-miR-151a-3p hsa-miR-378a-3p hsa-miR-140-3p	hsa-miR-148a-3p hsa-miR-151a-3p hsa-miR-378a-3p

## Data Availability

Data are contained within the article and Appendix A.

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
