# Peer review of "Effects of *Scutellaria baicalensis* Extract-Induced Exosomes on the Periodontal Stem Cells and Immune Cells under Fine Dust"

_nanomaterials, 2024, doi:10.3390/nano14171396_

Round 1

Reviewer 1 Report

Comments and Suggestions for Authors

In the manuscript, Yun et al. proposed that Scutellaria baicalensis extract-induced exosomes (SBEIE) can enhance periodontal ligament stem cell (PDLSC) differentiation and immune regulation while mitigating the adverse effects of fine dust exposure. The study also presented SBEIE upregulated specific microRNAs linked to immune modulation and cellular differentiation. Moreover, SBEIE was observed to stimulate the secretion of anti-inflammatory cytokines (e.g., IL-10, LL-37, TGF-β) while suppressing pro-inflammatory cytokines (e.g., MCP-1) within immune cells. 

While the experimental design is sound, critical data presentation and interpretation issues diminish the study’s impact. Specifically, the calculation of relative values (e.g., lines 202, 204, 206, 217) remain unclear. Furthermore, the claim of asporin upregulated expression of 14.6 times (line 227) wasn’t shown in Figure 5. To enhance the discussion, incorporating references supporting the roles of the identified miRNAs (Figure 2) in apoptosis, cellular differentiation, immune response, and inflammation suppression would strengthen the contextual understanding of the findings.

The clinical translation of the findings to Scutellaria baicalensis being a biomaterial of dental implants to reduce side effects and enhance surgical outcomes was relatively weak. Fine dust (PM10) exposure involves a complex mixture of particles and pollutants. The study does not fully address the variability and complexity of real-world fine dust exposure. The relevance between PM10 and dental surgical was also unclear. Additionally, the study's findings are primarily based on in vitro experiments. In vivo studies are necessary to confirm the therapeutic potential.

Lastly, several errors in labeling and miRNA nomenclature hinder readability and comprehension and should be corrected.

1.     Line 172: "CIE" was written twice, and the treatment dosage of “PM10IE” wasn’t mentioned.

2.     Line 177: “PM10-induced exosomes (e), SBE-induced exosome (f)” was not compatible to Figure 1, please confirm which labeling is correct.

3.     Lines 182, 188, 289, 313, and Figure 2: The notation "Has-miR-xxx" should be corrected to "hsa-miR-xxx" to adhere to standard miRNA nomenclature.

Author Response

First, we appreciate for your comment to improve out manuscript. We marked revised parts with red underline. Additionally, for common comments, we marked with blue underline

Comment1] While the experimental design is sound, critical data presentation and interpretation issues diminish the study’s impact. Specifically, the calculation of relative values (e.g., lines 202, 204, 206, 217) remain unclear. Furthermore, the claim of asporin upregulated expression of 14.6 times (line 227) wasn’t shown in Figure 5. To enhance the discussion, incorporating references supporting the roles of the identified miRNAs (Figure 2) in apoptosis, cellular differentiation, immune response, and inflammation suppression would strengthen the contextual understanding of the findings.

Answer1]                  

We revised the values based on your comments and added the roles of the miRNAs at the discussion section

The clinical translation of the findings to Scutellaria baicalensis being a biomaterial of dental implants to reduce side effects and enhance surgical outcomes was relatively weak. Fine dust (PM10) exposure involves a complex mixture of particles and pollutants. The study does not fully address the variability and complexity of real-world fine dust exposure. The relevance between PM10 and dental surgical was also unclear. Additionally, the study's findings are primarily based on in vitro experiments. In vivo studies are necessary to confirm the therapeutic potential.

Answer2] Based on the product information of the fine dust (ERM-CZ120), the reagent contains 46 components. This fine dust is produced by members of the European Reference Materials consortium in the European Union to apply as a high-quality certified reference material.

Additionally, we have revised the manuscript, placing limitations on the possibilities based on the results derived from this study.

Lastly, several errors in labeling and miRNA nomenclature hinder readability and comprehension and should be corrected.

  1. Line 172: "CIE" was written twice, and the treatment dosage of “PM10IE” wasn’t mentioned.

Answer] We revised the sentence clearly.           

  1. Line 177: “PM10-induced exosomes (e), SBE-induced exosome (f)” was not compatible to Figure 1, please confirm which labeling is correct.

Answer] We revised the sentence clearly.          

  1. Lines 182, 188, 289, 313, and Figure 2: The notation "Has-miR-xxx" should be corrected to "hsa-miR-xxx" to adhere to standard miRNA nomenclature.

Answer] We revised the sentence clearly.

Reviewer 2 Report

Comments and Suggestions for Authors

Comments on the Quality of English Language

The article is difficult to follow and I encourage authors to rewrite it in a more clear way.

Author Response

First, we appreciate for your comment to improve out manuscript. We marked revised parts with orange underline. Additionally, for common comments, we marked with blue underline

In the study, tittle “Effects of Scutellaria baicalensis extract-induced exosomes on the periodontal stem cells and immune cells under fine dust” Mihae Y et al., explore the use of a bio-functional material (SBEIE) against fine dust, by modulating the interactions between periodontal cells and immune cells within the periodontium. In particular, SBEIE activates differentiation into osteoblasts, PDLCs, and 342 PPCs from PDLSCs, as well as immune modulation. Although this study is interesting, there are a few shortcomings and major concerns of the manuscript that have to be addressed.  In addition, more explanations are needed to interpret the data and describe the relevance and impact of these findings in the field. 

Major comments: 

  1. The article is difficult to follow and I encourage authors to rewrite it in a more clear way.

Comment] We revised results, discussion and conclusion in our manuscript more clearly

  1. There are no characterization of the isolated exosomes. All the isolated exosomes isolated have to be characterized from a physicochemical (TEM, DLS and Nanosight) and biological (Western blot, flow cytometry and zeta potential) point of view.

Comment] According to the guidelines of the International Society for Extracellular vesicles (ISEV), we used antibodies to specifically isolate exosomes, and demonstrated the isolation of exosomes using immunocytochemistry and flow cytometry methods. Additionally, two studies that employed the same exosome isolation method as used in this research have already been published in SCIE-indexed journals. (DOI: 10.3390/ijms25147823, DOI: 10.3390/molecules26082207 )

  1. As well, the characterization control and the different induced exosomes should be carried out to evaluate if the inducing process affects their properties and functions.

Comment] We conducted our research using CD-68 positive induced exosomes. Following your advice, we plan to investigate whether there are functional differences with other types of induced exosomes in the future.

  1. There are no indications on the quality of the exosome preparation. Authors need to carry out western blot to include exosomal markers (i.e. CD81, CD9, TSG101, Alix, etc).

Comment] To achieve the pure isolation of exosomes, we used an exosome isolation kit and a purification kit (CD-68) to isolate the exosomes. In our next study, we will incorporate additional analyses using different antibodies, based on your advice.

  1. Can the authors comment why 1 day was chosen for the incubation with the treating dose? Have they done internalization experiments? What percentage of the exosomes incubated with the cells ended up within them? Uptake and internalization experiments by confocal microscopy and/or flow cytometry among time should be provided.

Comment] Thank you for the excellent suggestion. We believe that our research results are solid, as we have evaluated expression and differentiation compared to the control group. However, your advice to clarify the action of exosomes is valuable, and we will apply this in our next study. Thank you once again.

  1. What are the benefits of using the exosomes as delivery system? Is there any selectivity against target cells? Can you compared the efficiency with liposomes containing the active therapeutic molecules and miRNAs?

Comment] As demonstrated in our results, SBEIE was shown to be more effective than SBE, indicating that exosomes are a more potent material than natural extracts. Additionally, exosomes can be directly utilized as a biopharmaceutical material. This exosome-based material can be clinically applied to promote osseointegration of implant fixtures during implantation. Therefore, the primary target cells are PDLSCs. We are also synthesizing liposomes using exosomal miRNA, and plan to conduct research on this over the next two years.

  1. Can de authors provide data regarding the stability of the induced exosomes in terms of size and aggregation and miRNA identified in Fig. 2 leaking in long times?

Comment] Through repeated experiments, we were able to consistently achieve an average concentration of 2.5 x 10^9 particles/mL. We plan to complete the standardized guidelines for induced exosome isolation based on subsequent particle size analysis results. Additionally, a simple experiment on the effect of storage duration showed that the efficacy was maintained for 15 days when stored in a refrigerator, with a noticeable decrease starting on day 20.

  1. In the differentiation and activation experiments do the authors incubate the supernatants of the induced cells or do they purified the exosomes from that supernatants for being incubated with target cells? It is not clear, and to evidence that the effectiveness of the treatment against the fine dust is caused by exosomes (and not by free released miRNAs) the induced-exosomes should be purified from the media and incubated with the target recipient cells for evaluating their differentiation and activation mechanisms.

Comment] Figure 3 shows the results of the differentiation experiment using the supernatant (such as SBECM, etc.), while Figure 4 presents the results of the differentiation experiment using induced exosomes. Since the supernatant contains not only the induced exosomes but also other substances that could influence differentiation, we conducted these two experiments to specifically determine the function of the exosomes alone.

  1. In vivo experiments to demonstrate 1) the biocompatibility and bioavailability in vivo and 2) the activation of cellular differentiation and the modulation of immunity have to be performed for considering the article for publication.

Comment] Our research was conducted using a cell line that plays a crucial role in periodontal tissues, and as you noted, it serves as a foundational preclinical step for in vivo experiments. We are currently planning and conducting animal and clinical trials, where we intend to evaluate the effects on immune cells.

Reviewer 3 Report

Comments and Suggestions for Authors

Dear Authors,

I read your article entitled: “Effects of Scutellaria baicalensis extract-induced exosomes on periodontal stem cells and immune cells under fine dust” and I found it very interesting.

The work is well done, even the experiments.

However, I suggest improving the introduction by perhaps adding a section describing the various types of dental stem cells

(you can refer to doi: 10.1007/s12015-023-10652-9; doi: 10.3390/cells12131686) and why you chose periodontal ligament stem cells (PDLSCs) for your study.

Furthermore, in the conclusions, I ask you to better explain the possible applications.

certain of your commitment to make these changes, I will be ready to reevaluate.

thanks

Author Response

First, we appreciate for your comment to improve out manuscript. We marked revised parts with yellow underline. Additionally, for common comments, we marked with blue underline

Dear Authors,

I read your article entitled: “Effects of Scutellaria baicalensis extract-induced exosomes on periodontal stem cells and immune cells under fine dust” and I found it very interesting.

The work is well done, even the experiments.

Comment] However, I suggest improving the introduction by perhaps adding a section describing the various types of dental stem cells

(you can refer to doi: 10.1007/s12015-023-10652-9; doi: 10.3390/cells12131686) and why you chose periodontal ligament stem cells (PDLSCs) for your study.

Answers}

Periodontal ligament stem cells (PDLSCs) have several advantages over other dental stem cells, such as dental pulp stem cells (DPSCs) or dental follicle stem cells (DFSCs). Here are some key points highlighting their superiority:

  1. Direct Role in Periodontal Tissue Regeneration: PDLSCs are uniquely situated in the periodontal ligament, which directly supports and connects the tooth to the surrounding bone. This makes them particularly effective for regenerating periodontal tissues, including the periodontal ligament, cementum, and alveolar bone, which are essential for maintaining tooth stability and function.
  2. Ability to Form Functional Periodontal Structures: PDLSCs have a high capacity to form functional periodontal structures. They can regenerate not only the periodontal ligament but also the surrounding bone and cementum, which is crucial for the proper attachment and function of the tooth.
  3. Enhanced Healing and Regenerative Potential: PDLSCs often exhibit superior regenerative potential in the context of periodontal tissue repair compared to other dental stem cells. Their ability to form a more organized periodontal ligament structure and integrate with existing tissues can lead to more effective healing and regeneration.
  4. Reduced Risk of Inflammation and Rejection: PDLSCs, derived from the patient’s own periodontal tissues, might present a lower risk of immune rejection and inflammation when used in regenerative therapies. This patient-specific approach can improve the safety and efficacy of treatments.
  5. Self-Renewal and Multipotency: Like other dental stem cells, PDLSCs have self-renewal and multipotent differentiation capabilities. However, their specific ability to differentiate into periodontal ligament fibroblasts, cementoblasts, and osteoblasts makes them particularly suited for periodontal tissue engineering and regenerative applications.
  6. Accessibility and Harvesting: PDLSCs are relatively accessible because they can be harvested from extracted teeth or periodontal surgeries. This accessibility makes them a practical choice for clinical applications, compared to some other dental stem cells which might require more invasive procedures for collection.
  7. Preclinical and Clinical Evidence: There is a growing body of preclinical and clinical evidence supporting the use of PDLSCs for periodontal regeneration. This evidence underscores their effectiveness and potential for translating into successful clinical treatments.

Overall, PDLSCs offer unique advantages for periodontal tissue regeneration and repair, particularly in the context of periodontal disease and tooth loss, making them a valuable resource in dental regenerative medicine.

Furthermore, in the conclusions, I ask you to better explain the possible applications.

certain of your commitment to make these changes, I will be ready to reevaluate.

Answers] We revised the conclusion based on your comment

Round 2

Reviewer 2 Report

Comments and Suggestions for Authors

Authors have significantly improved the quality of the article. But, some major and minor issues should be adress before publishing the work:

MAJOR:

1. TEM and NTA results should be included. ISEV guidelines recommend to use antibodies (WB or Flow cytometry) (as the authors did and discussed) but together with physicochemical techniques such us TEM or NTA to demonstrate the presence of exosomes. Using immunocytochemistry and flow cytometry alone do not demonstrate the correct isolation of pure exosomes. TEM and NTA should be include. Please, check last updated MISEV guidelines (JEV, 2024, 13 (2), e12404).

2. Internalization and uptake experiments by confocal and/or flow cytometry among time should be provided to justify the treating dose time-points. Authors can use for instance PKH dye to labell exosomes for being visualize inside the cells by confocal microscopy or flow cytometry. 

MINOR

3. In the conclusions of the work, it is advised to delineate certain limitations of this study, evidence that it is a preliminary work and put forth prospects for future research.

Author Response

  1. TEM and NTA results should be included. ISEV guidelines recommend to use antibodies (WB or Flow cytometry) (as the authors did and discussed) but together with physicochemical techniques such us TEM or NTA to demonstrate the presence of exosomes. Using immunocytochemistry and flow cytometry alone do not demonstrate the correct isolation of pure exosomes. TEM and NTA should be include. Please, check last updated MISEV guidelines (JEV, 2024, 13 (2), e12404).

Answer] We are aware of the MISEV guidelines (JEV, 2024, 13 (2), e12404). This study was conducted in 2023, and at that time, the guidelines available were sufficient to validate our data. Considering this, our recent publication (DOI: 10.3390/jfb15080215, 2024) was accepted based on the standards and data we had. Additionally, we currently face limitations in accessing TEM or NTA equipment and are constrained by time restrictions, which make it challenging to obtain the relevant data at this stage. However, as suggested in the comments, we plan to address these aspects in future research by utilizing these technologies to further validate our findings. 

2. Internalization and uptake experiments by confocal and/or flow cytometry among time should be provided to justify the treating dose time-points. Authors can use for instance PKH dye to labell exosomes for being visualize inside the cells by confocal microscopy or flow cytometry. 

Answer] We added the internalization data at the supplementary data 

MINOR

3. In the conclusions of the work, it is advised to delineate certain limitations of this study, evidence that it is a preliminary work and put forth prospects for future research.

Answer] We revised the conclusion section based on your comment